

# Sea-ice thermodynamics can determine waterbelt scenarios for Snowball Earth

Johannes Hörner[1] and Aiko Voigt[1]

[1]Department of Meteorology and Geophysics, University of Vienna, Vienna, Austria

**Correspondence:** Johannes Hörner (johannes.hoerner@univie.ac.at)

**Abstract.** Snowball Earth refers to multiple periods in the Neoproterozoic during which geological evidence indicates that Earth was largely covered in ice. A Snowball Earth results from a runaway ice-albedo feedback, but there is an ongoing debate about how the feedback stopped: with fully ice-covered oceans or with a narrow strip of open water around the equator. The latter states are called waterbelt states and are an attractive explanation for Snowball Earth events because they provide a refugium for the survival of photosynthetic aquatic life, while still explaining Neoproterozoic geology. Waterbelt states can be stabilized by bare sea ice in the subtropical desert regions, which lowers the surface albedo and stops the runaway ice-albedo feedback. However, the choice of sea-ice model in climate simulations significantly impacts snow cover on ice and, consequently, surface albedo.

Here, we investigate the robustness of waterbelt states with respect to the thermodynamical representation of sea ice. We compare two thermodynamical sea-ice models, an idealized 0-layer Semtner model, in which sea ice is always in equilibrium with the atmosphere and ocean, and a 3-layer Winton model that is more sophisticated and takes into account the heat capacity of ice. We deploy the global climate model ICON-A in an idealized aquaplanet setup and calculate a comprehensive set of simulations to determine the extent of the waterbelt hysteresis. We find that the thermodynamic representation of sea ice strongly influences snow cover on sea ice over the range of all simulated climate states. Including heat capacity by using the 3-layer Winton model increases snow cover and enhances the ice-albedo feedback. The waterbelt hysteresis found for the 0-layer model disappears in the 3-layer model and no stable waterbelt states are found. This questions the relevance of a subtropical bare sea-ice region for waterbelt states and might help explain drastically varying model results on waterbelt states in the literature.

## 1 Introduction

The Neoproterozoic glaciations are commonly assumed to be of global extent and have become known under the term "Snowball Earth" (Kirschvink, 1992; Hoffman and Schrag, 2002). A Snowball Earth is initiated by a runaway ice-albedo feedback, where expanding ice, decreasing planetary albedo and falling global temperatures compose a feedback loop that leads to rapid glaciation until Earth is either fully ice-covered or the feedback is stopped by some other stabilizing mechanism (Pierrehumbert et al., 2011).





Climate states where the runaway ice-albedo feedback is stopped with a stable ice-border equatorward of 30° have been termed soft Snowball (Hyde et al., 2000; Yang et al., 2012a, b) or waterbelt states (Abbot et al., 2011; Rose, 2015; Brunetti et al., 2019; Ragon et al., 2022; Braun et al., 2022). For waterbelt states to be a feasible explanation for the Cryogenian glaciations, they have to be accessible from a temperate climate state and they have to be stable over a large range of atmospheric $CO_2$, as only then they can explain the long duration of the Snowball event and the survival of photosynthetic aquatic life (Hoffman
et al., 2017). Two major effects have been shown to allow states with low-latitude ice margins: Firstly, heat convergence at the sea-ice margin stopping ice growth thanks to heat convergence by the ocean circulation to ocean heat transport (Rose, 2015). Secondly, bare low-albedo sea ice in the subtropics stopping the ice-albedo feedback (Abbot et al., 2011). Here we focus on the latter effect, the Jormungand mechanism.

The Jormungand mechanism is based on the assumption that sea ice will not be covered by snow when it reaches the
subtropics, where the descending branch of the Hadley circulation results in net evaporation. Consequently, snow is melting and a larger region of darker bare sea ice is exposed, weakening the ice-albedo feedback until a stable state is reached. We refer to this region at the ice margin as the BAre Sea-Ice Region BASIR in this study. For BASIR to have an impact on the climate, a strong albedo contrast between dark bare sea ice and bright snow is required (Warren et al., 2002; Dadic et al., 2013). The width of BASIR then governs the strength of the Jormungand mechanism.

In this study, we assess the robustness of the Jormungand mechanism by examining sea ice and its snow cover in two different thermodynamical sea-ice models. That is, we ask: How robust is the Jormungand mechanism when using different sea-ice schemes? Recently, Braun et al. (2022) also postulated a strong influence of subtropical clouds on waterbelt stability, as the optical thickness of subtropical clouds determines the strength of the ice-albedo feedback when ice grows beneath the clouds. The stabilizing effect of clouds over ice is an overarching effect, as clouds mute the ice-albedo feedback. Here, however,
we isolate the effect of the sea-ice scheme on the ice-albedo feedback.

Abbot et al. (2010) showed that using more sea-ice layers generally reduces surface melting by reducing the "melt-ratchet" effect. This is because simple thermodynamic sea-ice models overestimate the daily and seasonal cycle of surface temperature and consequently overestimate surface melting (Semtner, 1984). For Snowball Earth initiation, using more layers results in stronger ice-albedo feedback because ice cover is increased and there is more snow on ice (Hörner et al., 2022). In this study,
we will extend the results of Hörner et al. (2022) and demonstrate that the waterbelt states that are stable and accessible with the simple 0-layer model are destabilized and absent when a more sophisticated 3-layer model is used.

The manuscript is structured as follows. In Section 2, the methods and the model setup are described. Section 3 presents the impacts of the different sea-ice schemes on the ice-albedo feedback and the waterbelt states. Section 4 discusses the implications for waterbelt states found previously in different climate models and provides a conclusion.





## 2 Methods

### 2.1 ICON-A

We deploy the ICOsahedral Non-hydrostatic Atmospheric model ICON-A (Giorgetta et al., 2018). There have been multiple studies on Snowball climate using the ICON model (e.g., Braun et al., 2022; Hörner et al., 2022; Ramme and Marotzke, 2022). Its predecessor ECHAM, which to a large part uses the same physics package (Giorgetta et al., 2018), has also been used in Snowball Earth simulations (e.g., Marotzke and Botzet, 2007; Voigt and Abbot, 2012; Abbot et al., 2012), including studies of waterbelt states (Abbot et al., 2011; Voigt et al., 2011).

The model and model setup are the same as in Hörner et al. (2022) and Braun et al. (2022). We use ICON-A with an effective horizontal resolution of 160 km and 45 vertical levels. ICON-A is coupled to a mixed-layer ocean of 50 m depth without any horizontal heat transport to simulate an aquaplanet without continents. A circular Kepler orbit and a solar constant reduced to 1285 $Wm^{-2}$ is applied to represent Neoproterozoic boundary conditions (Gough, 1981), as well as an idealized 360-day year. The albedo for cold snow and ice is 0.79 and 0.45, respectively. These values are linearly decreased to 0.66 for warm snow and 0.38 for warm ice, starting from 1 K below melting temperature. The albedo values are identical to those used in Braun et al. (2022), Hörner et al. (2022) and Abbot et al. (2011). Overall, the model setup follows Abbot et al. (2011), who first postulated the Jormungand state in idealized aquaplanet slab-ocean simulations with varying longwave forcing using the CAM and ECHAM models. Following Braun et al. (2022), we adapt the cloud scheme to allow ICON-A to simulate stable waterbelt states.

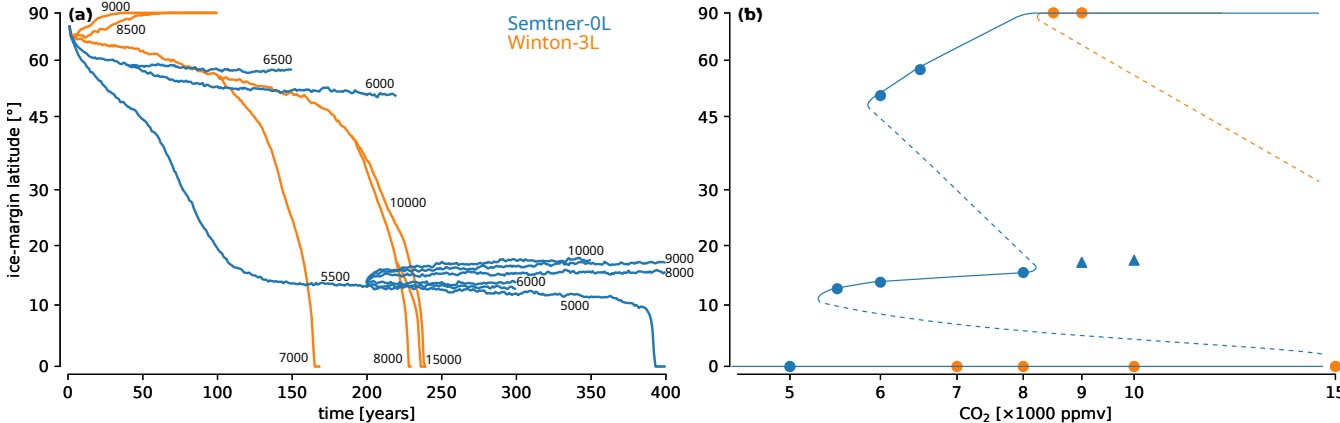

**Figure 1.** (a) Evolution of the yearly mean ice-margin latitude for simulations with Semtner-0L (blue) and Winton-3L (orange). Each line is labeled with its atmospheric $CO_2$ content in units of ppmv. (b) Estimated bifurcation diagrams. For equilibrated simulations, the ice-margin latitude is shown by solid circles. Simulations that continuously drift in snow cover are expected to melt and are depicted by upward-pointing triangles. Solid lines indicate the branches of stable equilibrium states, while dashed lines are for unstable equilibrium states. The exact positions of the tipping point and the branches of the unstable states are uncertain and should be considered a best guess.



We compare simulations with two different sea-ice schemes: the Semnter 0-layer scheme (Semtner, 1976), hereafter Semtner-0L, and the Winton 3-layer scheme (Winton, 2000), hereafter Winton-3L. Semtner-0L only considers surface temperature and does not predict internal ice temperatures. This means that the heat capacity of ice and brine pockets is neglected. Winton-3L
on the other hand takes into account the heat capacity. The model follows the energy-conserving approach of Bitz and Lipscomb (1999) and predicts ice temperatures at two internal levels and the surface. Energy stored in brine pockets is considered through a temperature-dependent enthalpy of fusion for ice. In the case of bare ice, Winton-3L allows a portion of incoming solar radiation to penetrate the ice. In both models, snow is converted to ice when the snow cover is submerged below the water level. For a more in-depth comparison of the ice models in the context of Snowball Earth initiation, we refer the reader
to Hörner et al. (2022).

ICON-A tends to become more unstable in colder climates. We therefore decrease the time step from the initial 10 min to 8 min and finally 6 min. All stable simulations on the temperate branch of the bifurcation diagram use a time step of 10 min, and a time step of 6 min on the waterbelt and Snowball Earth branches (cf. Fig. 1). As in Ramme and Marotzke (2022), we increase damping near the upper model boundary (Klemp et al., 2008) for time steps smaller than 10 min.

## 2.2   Simulation design

Two sets of simulations are performed, one using the Semnter-0L model (Semtner, 1976), the other using the Winton-3L model (Winton, 2000). Each simulation is characterized by its constant atmospheric $CO_2$ content and its initial condition. Most simulations are started from ice-free initial conditions, with sea-surface temperatures (SST) derived from AMIP runs of the present-day climate. The initial SST profile is zonally symmetric and symmetric with respect to the equator. The global
average SST is 291 K, ranging from 273 K at the poles to 301 K at the equator. Some simulations are branched off from other simulations after a certain ice-margin latitude has been reached. In these cases, a new simulation with a different constant atmospheric $CO_2$ content is started from a restart file of the parent simulation. This is done to find potentially hidden waterbelt states, to determine the width of the waterbelt hysteresis in the bifurcation diagram, and to save computing time.

## 3   Results

### 3.1   Simulation overview

Stable waterbelt states are found with the 0-layer Semtner sea-ice model but not the 3-layer Winton model. Figure 1 (a) shows the evolution of the ice margin for all simulations. All Winton-3L simulations either stay at an ice-free state or fall into a Snowball state in less than 250 years. Semtner-0L, in contrast, results in a waterbelt state when starting from ice-free initial conditions with a $CO_2$ content of 550 0ppmv. From there, more simulations are branched off with different $CO_2$ content to
assess the width of the waterbelt hysteresis.

A best guess of the bifurcation diagram is constructed from these simulations in Figure 1 (b). The waterbelt hysteresis in Semtner-0L extends over a range of $CO_2$ from 5500 to 8000 ppmv. The Semtner-0L bifurcation diagram is the same as in



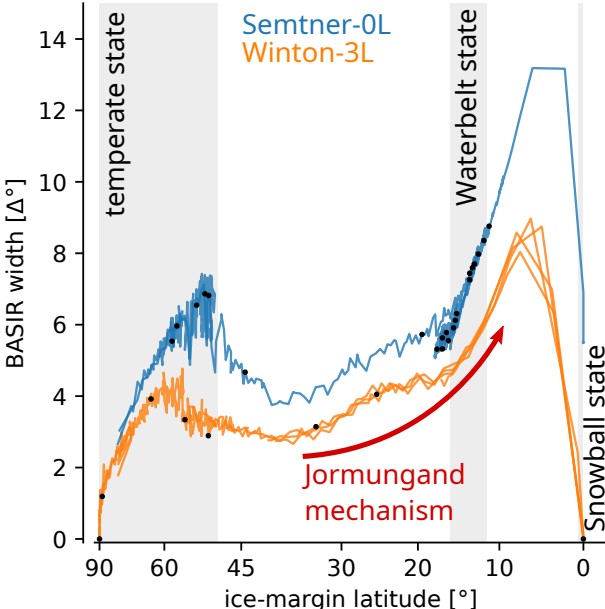

**Figure 2.** Width of the bare sea-ice region at the ice margin (BASIR) in deg latitude as a function of the latitude of the ice margin. For example, for an ice-margin latitude of 10° and a BASIR width of 10°, sea ice extends from the poles to 10° N/S, with bare ice between 10°-20° N/S and snow-covered ice between 20° and 90° N/S. Gray boxes indicate the regions where stable states are found (cf. Figure 1 b). Black points are plotted every 50 years after the start of each simulation to emphasize the position of stable states with a constant ice-margin latitude.

Braun et al. (2022) (their Figure 1a) since we are using the same model and model setup. In Winton-3L no waterbelt state is found up to a $CO_2$ value of 15000 ppmv. We cannot formally exclude the possibility of a hidden waterbelt state at even higher

values of $CO_2$ forcing. However, even if such a state existed, it would be geologically irrelevant because it would need to exist at a higher value of $CO_2$ than the tipping point of Snowball Earth initiation and would be inaccessible. Additionally, the strong ice-albedo feedback in Winton-3L for a low-latitude ice margin (see Subsection 3.4) makes such a state unlikely.

### 3.2   Jormungand mechanism

The difference between Semtner-0L and Winton-3L results from a strong difference in snow cover. Figure 2 shows the BASIR

width for all simulations as a function of the ice-margin latitude. Independent of $CO_2$, Semtner-0L results in a wider BASIR than Winton-3L. In Semtner-0L, BASIR widens with increasing global ice cover and reaches its maximum between 60° and 50°, near the tipping point marking the edge of the attractor of the temperate state. Following this, BASIR width sharply decreases as sea ice reaches the midlatitudinal precipitation maximum. When sea ice passes 30°, the Jormungand mechanism becomes apparent. BASIR width increases, as the negative P-E balance in the subtropics results in net snow melt. Ultimately,

this results in stable waterbelt states at an ice-margin latitude between 12° and 15°, with a BASIR width of around 6.5°.





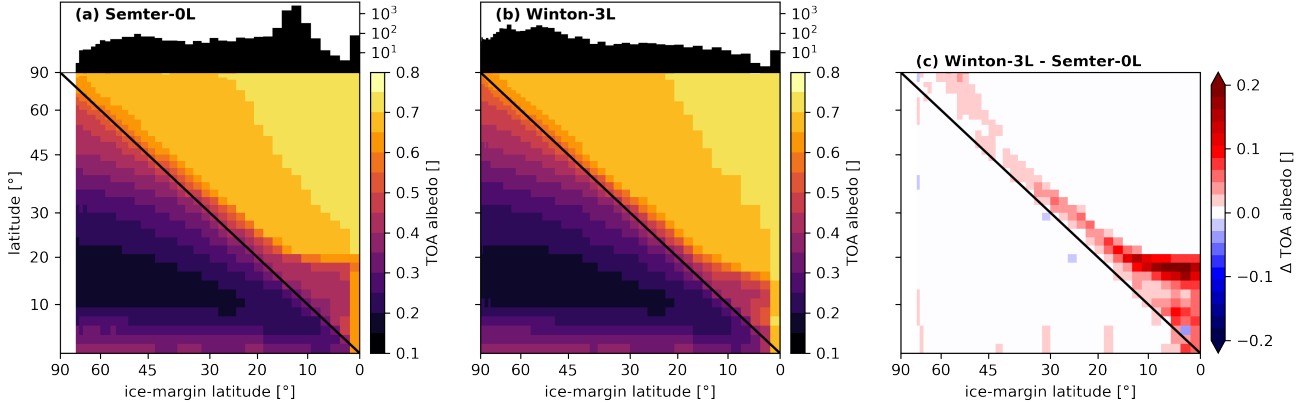

**Figure 3.** Meridional distribution (y-axis) of zonal-mean top-of-atmosphere (TOA) albedo (colors) as a function of ice-margin latitude (x-axis). The histograms on top show the number of data points per bin. (a) Semtner-0L. (b) Winton-3L. (c) Difference between Winton-3L and Semtner-0L.

In Winton-3L, the BASIR width shows the same general features, but BASIR is narrower for all sea-ice margins, as there is a larger snow cover on the ice. Although the Jormungand effect leads to a widening of BASIR when sea ice reaches the subtropics in Winton-3L as well, the effect is much smaller than in Semtner-0L. This explains why Winton-3L does not show any indications of stability in the region where Semtner-0L generates stable waterbelt states.

The difference in BASIR width modulates the ice-albedo feedback and hence determines waterbelt stability. To demonstrate this, we analyze the meridional structure of TOA albedo as a function of the ice-margin latitude. More specifically, we aggregated data from two transient simulations with each Semtner-0L and Winton-3L to cover the range from ice-free to Snowball states. For Semtner-0L, we aggregate the simulations with 5000 ppmv & 5500 ppmv $CO_2$, and for Winton-3L the siulations with 8000 ppmv & 8500 ppmv $CO_2$. Monthly & zonal mean output data of the simulations is binned according to the latitude

of the ice margin, and data in each ice-margin latitude bin is averaged over all months in that bin. The result is the meridional distribution of TOA albedo as a function of ice-margin latitude shown in Figure 3. The number of values per bin varies significantly, depending on how long the ice margin stays in that bin. This is illustrated by the histograms on top of Figure 3)

The differences in BASIR width at the ice margin result in different TOA albedos in Semtner-0L compared to Winton-3L (Figure 3 (c). The largest difference is located directly poleward of the ice margin. The albedo difference around the ice

margin increases strongly as soon as the ice margin moves beyond 30°. This indicates the larger BASIR widening and thus stronger Jormungand mechanism in Semtner-0L. Importantly, strong differences in the TOA albedo occur for subtropical sea ice equatorward of 20°. This is because subtropical sea ice is bare in Semtner-0L as demanded by the Jormungand mechanism, but some snow in fact is able to persist on top of subtropical sea ice in Winton-3L. Consequently, waterbelt states are found in Semtner-0L with stable ice margins between 15 and 20°, and no waterbelt states are found in Winton-3L.





### 3.3 TOA energy budget

The widening of the BASIR when ice enters the subtropics lowers the TOA albedo and leads to a weakening of the ice-albedo feedback. In Semtner-0L, but not in Winton-3L, the weakening is strong enough to bring the TOA energy budget into equilibrium for waterbelt states. This behavior is illustrated in Figure 4, which shows the global mean TOA energy budget in panel (a) and its shortwave and longwave components in panels (b) and (c).

Both ice models start out with a TOA energy budget close to equilibrium for ice margins corresponding to temperate climate states. When ice advances equatorward of $50°$, the energy budget becomes more and more negative due to the runaway ice-albedo feedback. For Semtner-0L, the Jormungand mechanism becomes evident as the TOA energy budget reaches a local minimum for an ice margin of $30°$, and subsequently increases again until equilibrium is reached in a waterbelt state. Instead, for Winton-3L the TOA energy budget only shows a slight slowdown and the Jormungand mechanism is not strong enough to restore the TOA energy budget into equilibrium.

The longwave component of the TOA energy budget depicted in Figure 4 (c) is mostly determined by temperature and the Planck feedback, and hence essentially the same in Semtner-0L and Winton-3L. Small differences are due to the different atmospheric $CO_2$ content in the simulations. For temperate states, Winton-3L emits marginally less longwave radiation compared to Semtner-0L since a larger $CO_2$ content is used in the Winton-3L simulations. In the region of the Jormungand state, Semtner-0L and Winton-3L have similar values of atmospheric $CO_2$ (see Figure 1 (b), and hence a very similar longwave radiative flux.

The diverging evolution of the TOA energy budget between Semtner-0L and Winton-3L is driven by the shortwave component, i.e., the ice albedo. The global mean of the absorbed shortwave radiative flux as a function of ice-margin latitude is depicted in Figure 4 (b). The slope of this line measures the strength of the ice-albedo feedback. The amount of absorbed shortwave radiation slowly diverges between the two ice models as sea ice expands from the midlatitudes into the subtropics, with Semnter-0L absorbing more shortwave radiation. This is a consequence of the increasing difference in surface albedo due to the difference in BASIR width shown in Figure 3. In the waterbelt state, the difference is up to $2.5 \ \mathrm{Wm}^{-2}$ and increases further for ice margins closer to the equator, as the difference in BASIR increases. The difference between Semtner-0L and Winton-3L is not driven by clouds, as the clear-sky energy budget exhibits the same behavior (not shown).

In summary, the larger surface melting in Semtner-0L due to the stronger melt-ratchet effect results in less snow on the ice and a wider BASIR. This weakens the ice-albedo feedback and strengthens the Jormungand mechanism by decreasing surface and TOA albedo near the sea-ice margin. Consequently, in Semtner-0L the TOA energy budget can be in equilibrium for low-latitude sea-ice margins and waterbelt states are possible, whereas Winton-3L falls into a Snowball state.

### 3.4 Effect of ice model on snow cover

To further illustrate that Winton-3L favors more snow on ice, we perform an additional simulation with Winton-3L that is restarted from a stable waterbelt state simulated with Semtner-0L. The Winton-3L simulation is branched off from year 400 of the Semtner-0L simulation with 8000 $\mathrm{ppmv}$ $CO_2$ at year 400 (cf. Figure 1). The initial ice internal temperatures that are





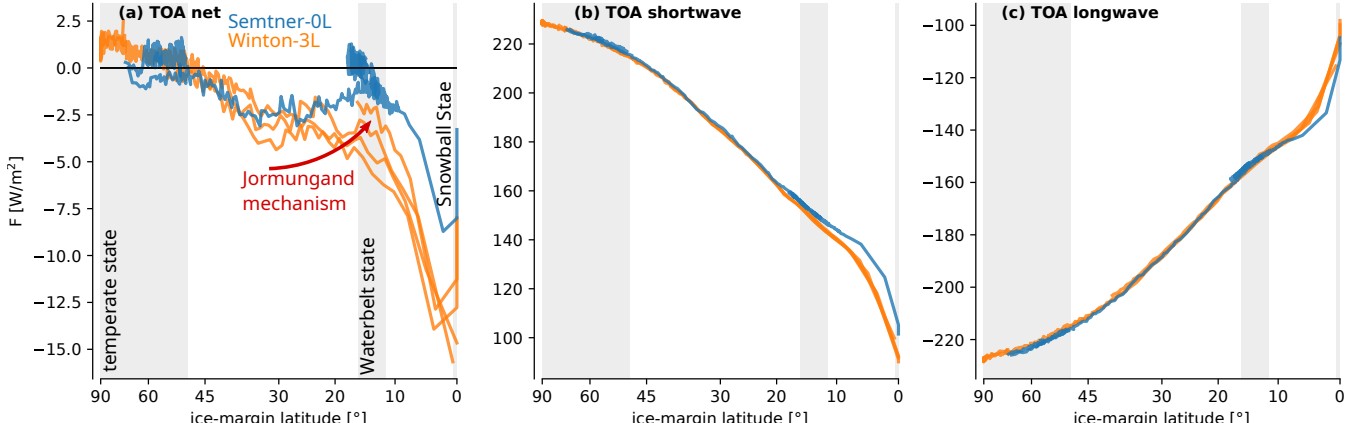

**Figure 4.** Components of the global-mean top-of-atmosphere (TOA) energy budget as a function of the ice-margin latitude for all simulations with Semtner-0L (blue) and Winton-3L (orange). (a) Net energy budget, (b) absorbed shortwave radiation, (c) outgoing longwave radiation. Fluxes are defined as positive downward for all components. Gray boxes indicate the regions where stable states were found.

required for the Winton-3L model are obtained from the ice surface temperatures from the restart file of the Semtner-0L simulation.

The 3L-Winton simulation falls into a Snowball state after 6 years. This is shown in terms of surface albedo in Figure 5. The lower part of the figure depicts the first year in higher temporal resolution, and the upper part the subsequent years until global glaciation is achieved. In Winton-3L surface albedo increases quickly within the first few months in the region of ice that is bare in Semtner-0L. This indicates that snow quickly accumulates on ice in Winton-3L and the BASIR narrows. After 6 months the border between snow-free and snow-covered ice has clearly shifted towards the equator in Winton-3L (dotted lines), and

the increase in surface albedo has led to global cooling and more ice growth also in the summer (northern) hemisphere. In the following months and years until glaciation, this poses a feedback loop of increased snow cover in both summer and winter hemispheres, larger ice cover in the summer hemisphere, higher surface albedo, and lower temperatures.

## 4    Conclusions

Previous work proposed that waterbelt states with low-latitude ice margins are possible thanks to the Jormungand mechanism.

The Jormungand mechanism is based on a widening of the region of bare sea ice (abbreviated as BASIR) near the sea-ice margin when sea ice enters the subtropics, which decreases subtropical ice albedo and can stop the runaway ice-albedo feedback and stabilize the ice margin at around 15° (Abbot et al., 2011). In this study, we investigate the robustness of such waterbelt states with respect to the representation of the thermodynamics of sea ice. To this end, we conduct simulations with the ICON-A atmosphere model in idealized aquaplanet setup using the simple 0-layer Semtner sea-ice model and the more sophisticated

energy-conserving 3-layer Winton model.



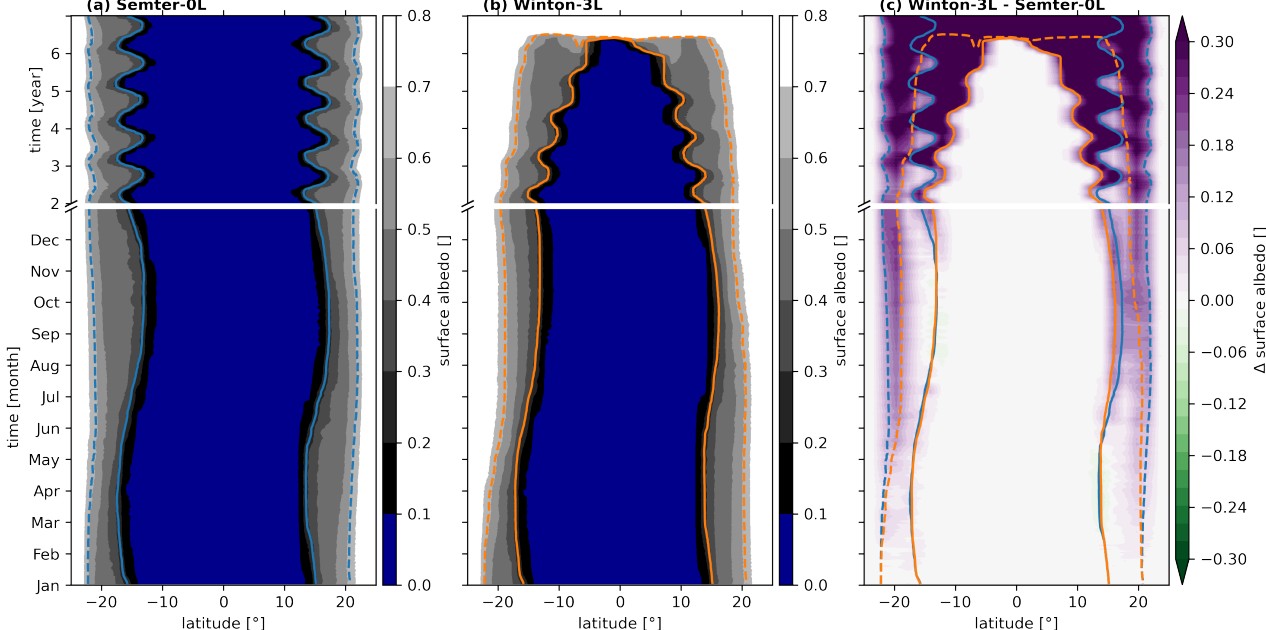

**Figure 5.** Zonal-mean surface albedo for a Semnter-0L (a) and a Winton-3L simulation (b) as a function of time. The Winton-3L simulation is restarted from the stable Semtner-0L waterbelt simulation. Both simulations use 8000 ppmv $CO_2$. Solid lines indicate the ice margin, and dashed lines indicate the snow margin. (c) Difference in surface albedo between the Winton-3L and Semtner-0L simulations, with the ice and snow margin lines of both simulations overlaid. The initial year is shown with an enlarged time axis, the following 5 years are depicted with a squeezed time axis.

The main finding of our work is that stable waterbelt states are possible and accessible in the 0-layer Semtner model but not in the 3-layer Winton model. We show that the lack of waterbelt states in the 3-layer Winton model results from a narrower BASIR, which weakens the Jormungand mechanism and strengthens the ice-albedo feedback compared to the simulations with the 0-layer Semtner model. The BASIR is narrower in the 3-layer Winton model because taking into account the vertical

resolution and ice heat capacity decreases surface melting and allows more snow to accumulate on top of sea ice, in particular in the subtropics.

That ice vertical resolution is important to correctly simulate surface melting and hence Snowball Earth climates have been shown before by Abbot et al. (2010), Yang et al. (2012a, b) and Hörner et al. (2022), who demonstrated that insufficient vertical resolution overestimates surface melting due to a melt-ratchet effect. In Hörner et al. (2022) we specifically showed

that insufficient vertical resolution enhances Snowball Earth initiation by allowing more ice and snow to persist into the summer months. Here, we study the effect of ice vertical resolution on the ice-albedo feedback over a wider range of climate states and find that the impact of vertical resolution becomes increasingly important as the ice margin moves toward the equator. As a



result, the vertical resolution of sea ice can determine the existence and stability of a waterbelt state by controlling the width of the BASIR.

Our results question the validity of waterbelt states. However, previous studies with a range of different sea ice models found waterbelt states. So how important is the Jormungand mechanism and how does it compare to other mechanisms? We briefly discuss this question in the following and note that although the Jormungand mechanism may not be able to stabilise a waterbelt state on its own, it might add to other stabilizing mechanisms so that a waterbelt state might be possible.

– Multiple studies found waterbelt states in the atmospheric model CAM and the earth system model family CCSM/CESM
that CAM is part of (Abbot et al., 2011; Yang et al., 2012a, b; Wolf et al., 2017; Braun et al., 2022). These studies used a 5-layer sea-ice scheme, which based on our results should lead to an even narrower BASIR and weaker Jormungand mechanism than in our 3-layer Winton simulations. Braun et al. (2022) argued that optically thick mixed-phased clouds in the subtropics are in fact needed to stabilize the waterbelt state by muting the ice-albedo feedback in the CAM idealized aquaplanet simulations of Abbot et al. (2011). On the other hand, it seems likely that the stable waterbelt states simulated
in the coupled simulations of Yang et al. (2012a) and Yang et al. (2012b) are possible thanks to heat convergence near the ice margin by the ocean circulation.

– The Massachusetts Institute of Technology General Circulation Model MITgcm (e.g., Marshall et al., 2004) allows for waterbelt states that are solely stabilized by ocean heat transport (e.g., Rose, 2015; Brunetti et al., 2019; Ragon et al., 2022; Zhu and Rose, 2022). MITgcm uses the same thermodynamic 3-layer Winton sea-ice model as our study. The
waterbelt state in these studies does not show any BASIR. According to Rose (2015), this is because the ice margin is located further poleward, between 21° and 30° latitude. This is outside of the influence of the descending branch of the Hadley cell and thus outside of the influence of the Jormungand mechanism. As they are stabilized by different mechanisms, they can be considered different climate states. This demonstrates that both mechanisms can stabilize a waterbelt state on their own.

– The atmosphere model ECHAM5 was used by Abbot et al. (2011) to first demonstrate the Jormungand mechanism in idealized aquaplanet simulation, and its coupled atmosphere-ocean version ECHAM5-MPIOM was used by Voigt and Abbot (2012) to test the robustness with respect to sea-ice dynamics. Both ECHAM5 and ECHAM5-MPIOM used a 0-layer Semtner model. ECHAM5 did not track snow on the ice, and because of this, the BASIR had to be hard-coded into the model. The ECHAM5-MPIOM simulations showed that while the Jormungand mechanism appeared to be active,
it was not strong enough to allow for waterbelt states that are separated from temperate states by a bifurcation and the strong sea ice dynamics in the model inhibited stable subtropical ice margins.

– The ICON model is the successor of ECHAM5 and ECHM5-MPIOM and also uses the 0-layer Semtner sea-ice scheme in its standard version. Braun et al. (2022) demonstrated that the ICON model has difficulties in simulating stable waterbelt states because subtropical clouds are not reflective enough in the model. If cloud optical thickness and cloud reflectivity
are increased, the Jormungand mechanism leads to waterbelt states in ICON.



Our results show that in order for the Jormungand mechanism to lead to stable waterbelt states, the region of bare sea ice needs to be sufficiently wide. Simulations with simplified sea-ice models overestimate the width of the bare sea-ice region and therefore likely overestimate the stability of waterbelt states. Consequently, waterbelt states that rely exclusively on the Jormungand mechanism might only be possible for rather peculiar combinations of a coarse vertical resolution of ice and 235 optically thick subtropical clouds. This suggests that the Jormungand mechanism itself might be secondary and robust waterbelt states are only possible if other stabilising mechanisms, particularly that from ocean heat transport, are also active.

*Code and data availability.* Code for simulation runs, postprocessing and creation of figures is available at https://gitlab.phaidra.org/climate/ hoerner-voigt-waterbelt-seaice-esd2023-submitted. Data required to create the figures of this manuscript is available at https://ucloud.univie. ac.at/index.php/s/smdVW02jXICISBv. These sources are only available as long as the manuscript is under revision. Once the manuscript 240 has been accepted for publication, the final data will be available at https://doi.org/10.25365/phaidra.429.

*Author contributions.* JH performed the simulations and analysis and wrote the initial draft. JH and AV wrote the final draft. AV supervised the project.

*Competing interests.* The authors declare no competing interests.

*Acknowledgements.* Open Access funding provided by University of Vienna. The computational results presented have been achieved using 245 the Vienna Scientific Cluster (VSC). AI tools have been used to improve the grammar, writing style, and wording for parts of this manuscript.





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
