# Peer review of "Sea-ice thermodynamics can determine waterbelt scenarios for Snowball Earth"

_EGUsphere, 2023_

## Author Comment (AC1)

**Sea-ice thermodynamics can determine waterbelt scenarios for Snowball Earth - Response to Reviewers**

Johannes Hörner          Aiko Voigt

December 14, 2023

We would like to thank the reviewers for their constructive feedback. Below we list our responses to the reviewers' comments and how we plan to incorporate them into the manuscript. We believe that revising our manuscript according to the reviewers' comments will greatly improve the manuscript and hope that the resulting manuscript will be accepted for publication.

**RC1**

**Major comments**

1. **On the modern Earth, snow and ice occur mostly in the polar regions, where the Sun is low, or in midlatitudes in winter, when the days are short, so snow and ice do not get exposed to much solar energy. The exact values of snow and ice albedo become much more important to global climate when ice advances into low latitudes (Snowball Earth), but many snowball modelers have been using inappropriate albedos. On line 66, the albedo for cold snow should be 0.83, not 0.79. Warm (melting) snow has albedo 0.76 if it is clean, not 0.66 (these values are in Table 2 of Webster and Warren, 2022; WW22), but the value 0.66 used by HV could be appropriate if the snow contains dust. Bare cold thick sea ice has albedo 0.47-0.49, slightly higher than HV's 0.45 (Table 1 of Warren et al., 2002). Melting Arctic sea ice develops a granular surface scattering layer (SSL), resulting in a relatively high albedo of 0.60 (WW22), i.e. higher than that of bare cold thick sea ice. In depressions on the melting Arctic sea ice, little ponds form, with albedo 0.2 (WW22). So on line 67, the albedo 0.38 used by HV for warm ice implies that the areal fraction of ponds on melting ice is 55%, which is far more than is now found on the Arctic Ocean in summer (Table 1 of WW22).**

   **In summary, the albedos used by HV are lower than observed. Ideally HV would rerun their models with more-appropriate albedos. If they choose not to make this revision, their paper is still a useful test of their hypothesis, since they are using the same values in both models. But the authors should acknowledge that the albedos they used are too low. This acknowledgment will strengthen their conclusion, because using higher, more-realistic, albedos for snow and ice in models will make the waterbelt even less accessible.**

   Thank you for this detailed comment. We agree that the albedo values we are using are indeed rather on the low side. We chose these idealized values over more realistic ones to provide comparability with other model studies on waterbelt states. Exactly the same values for albedo are used in the paper that first postulated the Jormungand hypothesis Abbot et al. (2011), as well as more recent work (Braun et al., 2022a,b; Hörner et al., 2022). Especially the comparison with Braun et al. (2022a) and Hörner et al. (2022) is beneficial, as these studies use the same setup. Rerunning our models with more realistic albedos would require an immense amount of computing (and real) time, that's why we do not intend do this in the context of this study. We will include this information into the manuscript.

**Minor comments**

1. **Figure 1a. I'm glad you're showing the time scale of the transition to global glaciation, and also on Figure 5. This is interesting. If you had an interactive deep ocean, I assume the time required to reach the snowball state would be much longer.**

   Thank you for this comment. Indeed, compared to model studies using an interactive ocean, global glaciation happens much faster here. We also observe this fast glaciation when using other models with a slab ocean, e.g CAM in Braun et al. (2022a).

2. **Figure 2 is confusing. The caption says "Gray boxes indicate the region where stable states are found". Since the orange Winton curves do not avoid the gray box labelled "Waterbelt state", it seems to show that a stable waterbelt is accessible with the Winton model, in conflict with the message of the paper. Some changes to the figure and/or the caption are needed to clear up this confusion. Maybe the figure is showing the non-equilibrium time-dependent path of transition into a snowball, rather than the equilibrium stable state?**

   Thanks for this point, this is indeed misleading. The figure shows the bare sea-ice region width as a function of ice-margin latitude for all simulations. It thus includes both transient states as well as stable states. As this figure alone does not allow to identify stable states, we included the gray boxes that indicate the regions of ice-margin latitude where stable states are found using the Semtner-0L model. With this we want to highlight the fact that there are indeed no stable waterbelt states found with the Winton-3L model in the region where we would expect them from the results with the Semtner-0L model. We will point this out in the figure by changing the label of the gray box and add more details in the caption.

3. **Line 106. "a higher value of CO2 than the tipping point of Snowball Earth initiation". What is the CO2 value for the tipping point?**

   In the Semtner-0L simulations of this study the tipping point is between 8000ppmv and 8500ppmv, we will include this in the manuscript.

4. **Figure 4a. Why does TOAnet not go to zero in the snowball state (ice-margin latitude zero)?**

   We stopped all simulations shortly after they reached the Snowball state (see Figure 1a in the original manuscript). All of these simulations are not in equilibrium yet, TOAnet is still negative and global mean surface temperature is still decreasing. Reaching TOAnet=0 would require the simulations to run an estimated 10-100 years longer. We did not do this as the Snowball climate is not the focus of this study. Additionally, the ICON model becomes increasingly unstable in the Snowball state.

5. **Line 22. Change "decreasing" to "increasing".**

6. **Line 31. "Thanks to heat convergence by the ocean circulation to ocean heat transport" is awkwardly worded. Maybe the last four words can be deleted.**

7. **Line 35. Change "evaporation" to "sublimation/evaporation".**

8. **Line 63. Change "45 vertical levels" to "45 atmospheric vertical levels". Otherwise the reader may think you have 45 levels in the sea ice.**

9. **Line 84. Change "the upper model boundary" to "the upper boundary of the model's atmosphere".**

10. **Line 86. Change Semnter to Semtner.**

11. **Line 99. Change "550 0ppmv" to "5500 ppmv".**

12. **Figure 3. In the labels at the top of parts a and c, change Semter to Semtner.**

13. **Line 123. Change "siulations" to simulations**

14. **Line 130. Change "beyond 30" to "below 30"**

15. **Figure 4 caption line 1. Change "the ice-margin latitude" to "the transient ice-margin latitude" because TOAnet $\neq$ 0 means we're not in equilibrium.**

16. **Figure 5. In the labels at the tops of parts a and c, change Semter to Semtner.**

17. **Line 299. Change Marnoun to Marinoan.**

We agree with all other minor comments, we will correct these in the manuscript. Thank you for pointing out the typographical errors and ambiguities in the wording.

**RC2**

**Major comments**

1. **The paper spends considerable time discussing the response of bare sea-ice region at the ice margin on Semtner-0L and Winton-3L. However, the discussion on the net evaporation due to the Hadley cell in 10°-20°N/S is missing. Is the net evaporation the same in that region in both sea ice configurations? The statement in Line 189-191 needs a support from some energy budget analysis to estimate the influence of the ice heat capacity more quantitatively.**

   Thank you for this comment, we believe that including this will greatly improve our manuscript. We will add an additional figure to the manuscript that displays the zonal mean precipitation, evaporation, P-E and surface ablation of sea-ice for both Semtner-0L and Winton-3L (Figure 1a in this document). P-E is almost identical in both models, both show a negative P-E balance near the sea-ice margin. Semtner-0L shows a slightly more negative P-E balance near the sea-ice margin (maximum difference of 0.09m/year). Additionally, Semtner-0L shows slightly more precipitation near the sea-ice margin.

   Concerning surface ablation, Semtner-0L shows much larger values (maximum difference of 1.1m/year). This is the melt-ratchet effect, the periodical melting of sea ice and snow at the surface, which is larger in the Semtner-0L model due to the neglected heat capacity. The larger surface melting in Semnter-0L is partially compensated by larger ice growth at the bottom (also see Hörner et al. 2022). But the larger ice growth at the bottom cannot compensate the surface melting of snow. Due to this, a larger cycle of melting and freezing is more efficient in decreasing snow thickness than it is in decreasing ice thickness. This results in less snow thickness in Semtner-0L (Figure 1b in this document) and consequently a larger bare sea-ice region. We will add additional explanation of this effect into the manuscript.

**Minor comments**

1. **In Line 25-33, a previous study showed that the surface topography could facilitate the formation of waterbelt solution by reducing the snow coverage over continental interior when climate became cooler and cooler (Liu et al., 2018; DOI: 10.1175/JCLI-D-17-0821.1).**

   Thank you for this comment, this study is a useful addition to the importance of snow cover dynamics for waterbelt states, we will discuss this in the conclusion section of the manuscript.

[Figure]

Figure 1: a) Zonal mean precipitation, evaporation, precipitation-evaporation, as well as ice surface ablation. b) Zonal mean snow thickness on sea ice. The data is processed equivalent to Figure 3 in the original manuscript. All months with an ice-margin latitude between $11.25°$ and $13.125°$ are selected from the simulations with 5000ppmv and 5500ppmv $CO_2$ for Semnter-0L, and with 8000ppmv and 8500ppmv $CO_2$ for Winton-3L. The center of the range of ice-margin latitudes is indicated by gray vertical lines.

**References**

Abbot, D. S., Voigt, A., and Koll, D. (2011). The Jormungand global climate state and implications for Neoproterozoic glaciations. *J. Geophys. Res.*, 116(D18103).

Braun, C., Hörner, J., Voigt, A., and Pinto, J. G. (2022a). Ice-free tropical waterbelt for Snowball Earth events questioned by uncertain clouds. *Nat. Geosci.*, 15(6):489–493.

Braun, C., Voigt, A., Hoose, C., Ekman, A. M. L., and Pinto, J. G. (2022b). Controls on Subtropical Cloud Reflectivity during a Waterbelt Scenario for the Cryogenian Glaciations. *Journal of Climate*, 35(21):3457–3476.

Hörner, J., Voigt, A., and Braun, C. (2022). Snowball Earth Initiation and the Thermodynamics of Sea Ice. *J. Adv. Model. Earth Syst.*, 14(8):e2021MS002734.